

# Geometric-Based Pruning Rules for Change Point Detection in Multiple Independent Time Series

ISSN 2824-7795

Liudmila Pishchagina[1]    Universit'e Paris-Saclay, CNRS, Univ Evry, Laboratoire de Math'ematiques et Mod'elisation d'Evry, 91037, Evry-Courcouronnes, France

Guillem Rigaill    Universit'e Paris-Saclay, CNRS, Univ Evry, Laboratoire de Math'ematiques et Mod'elisation d'Evry, 91037, Evry-Courcouronnes, France

Universit'e Paris-Saclay, CNRS, INRAE, Univ Evry, Institute of Plant Sciences Paris-Saclay (IPS2), Orsay, France

Vincent Runge    Universit'e Paris-Saclay, CNRS, Univ Evry, Laboratoire de Math'ematiques et Mod'elisation d'Evry, 91037, Evry-Courcouronnes, France

Date published: 2023-06-05    Last modified: 2024-02-19

**Abstract**

We address the challenge of identifying multiple change points in a group of independent time series, assuming these change points occur simultaneously in all series and their number is unknown. The search for the best segmentation can be expressed as a minimization problem over a given cost function. We focus on dynamic programming algorithms that solve this problem exactly. When the number of changes is proportional to data length, an inequality-based pruning rule encoded in the PELT algorithm leads to a linear time complexity. Another type of pruning, called functional pruning, gives a close-to-linear time complexity whatever the number of changes, but only for the analysis of univariate time series. We propose a few extensions of functional pruning for multiple independent time series based on the use of simple geometric shapes (balls and hyperrectangles). We focus on the Gaussian case, but some of our rules can be easily extended to the exponential family. In a simulation study we compare the computational efficiency of different geometric-based pruning rules. We show that for a small number of time series some of them ran significantly faster than inequality-based approaches in particular when the underlying number of changes is small compared to the data length.

*Keywords:* multivariate time series, multiple change point detection, dynamic programming, functional pruning, computational geometry

# Contents

[1]Corresponding author: liudmila.pishchagina@univ-evry.fr

# Introduction

A National Research Council report (Data et al. 2013) has identified change point detection as one of the "inferential giants" in massive data analysis. Detecting change points, either a posteriori or online, is important in areas as diverse as bioinformatics (Olshen et al. 2004; Picard et al. 2005), econometrics (Bai and Perron 2003; Aue et al. 2006), medicine (Bosc et al. 2003; Staudacher et al. 2005; Malladi, Kalamangalam, and Aazhang 2013), climate and oceanography (Reeves et al. 2007; Ducré-Robitaille, Vincent, and Boulet 2003; Killick, Fearnhead, and Eckley 2012; Naoki and Kurths 2010), finance (Andreou and Ghysels 2002; Fryzlewicz 2014), autonomous driving (Galceran et al. 2017), entertainment (Rybach et al. 2009; Radke et al. 2005; Davis, Lee, and Rodriguez-Yam 2006), computer vision (Ranganathan 2012) or neuroscience (Jewell, Fearnhead, and Witten 2019). The most common and prototypical change point detection problem is that of detecting changes in mean of a univariate Gaussian signal and a large number of approaches have been proposed to perform this task (see among many others (Yao 1984; Lebarbier 2005; Harchaoui and Lévy-Leduc 2010; Frick, Munk, and Sieling 2013; Anastasiou and Fryzlewicz 2022) and the reviews (Truong, Oudre, and Vayatis 2020; Aminikhanghahi and Cook 2017)).

*Penalized cost methods.* Some of these methods optimize a penalized cost function (see for example (Lebarbier 2005; Auger and Lawrence 1989; Jackson et al. 2005; Killick, Fearnhead, and Eckley 2012; Rigaill 2015; Maidstone et al. 2017). These methods have good statistical guarantees (Yao 1984; Lavielle and Moulines 2000; Lebarbier 2005) and have shown good performances in benchmark simulation (Fearnhead, Maidstone, and Letchford 2018) and on many applications (Lai et al. 2005; Liehrmann, Rigaill, and Hocking 2021). From a computational perspective, they rely on dynamic programming algorithms that are at worst quadratic in the size of the data, *n.* However using inequality-based and functional pruning techniques (Rigaill 2015; Killick, Fearnhead, and Eckley 2012; Maidstone et al. 2017) the average run times are typically much smaller allowing to process

very large profiles ($n > 10^5$) in a matter of seconds or minutes. In detail, for one time series:

- if the number of change points is proportional to $n$ both PELT (Killick, Fearnhead, and Eckley 2012) (a version of OP which uses inequality-based pruning) and FPOP (Maidstone et al. 2017) (a version of OP which uses functional pruning as in (Rigaill 2015)) are on average linear (Killick, Fearnhead, and Eckley 2012; Maidstone et al. 2017);
- if the number of change points is fixed, FPOP is quasi-linear (on simulations) while PELT is quadratic (Maidstone et al. 2017).

*Multivariate extensions.* This paper focuses on identifying multiple change points in a multivariate independent time series. We assume that changes occur simultaneously in all dimensions, their number is unknown, and the cost function or log-likelihood of a segment (denoted as $\mathscr{C}$) can be expressed as a sum across all dimensions $p$. Informally, that is,

$$\mathscr{C}(segment) = \sum_{k=1}^{p} \mathscr{C}(segment, \text{ time series } k).$$

In this context, the PELT algorithm can easily be extended for multiple time series. However, as for the univariate case, it will be algorithmically efficient only if the number of non-negligible change points is comparable to $n$. In this paper, we study the extension of functional pruning techniques (and more specifically FPOP) to the multivariate case.

At each iteration, FPOP updates the set of parameter values for which a change position $\tau$ is optimal. As soon as this set is empty the change is pruned. For univariate time series, this set is a union of intervals in $\mathbb{R}$. For parametric multivariate models, this set is equal to the intersection and difference of convex sets in $\mathbb{R}^p$ (Runge 2020). It is typically non-convex, hard to update, and deciding whether it is empty or not is not straightforward.

In this work, we present a new algorithm, called Geometric Functional Pruning Optimal Partitioning (GeomFPOP). The idea of our method consists in approximating the sets that are updated at each iteration of FPOP using simpler geometric shapes. Their simplicity of description and simple updating allow for a quick emptiness test.

The paper has the following structure. In Section 1 we introduce the penalized optimization problem for segmented multivariate time series in case where the number of changes is unknown. We then review the existing pruned dynamic programming methods for solving this problem. We define the geometric problem that occurs when using functional pruning. The new method, called GeomFPOP, is described in Section 2 and based on approximating intersection and exclusion set operators. In Section 3 we introduce two approximation types (sphere-like and rectangle-like) and define the approximation operators for each of them. We then compare in Section 4 the empirical efficiency of GeomFPOP with PELT on simulated data.

# 1 Functional Pruning for Multiple Time Series

## 1.1 Model and Cost

We consider the problem of change point detection in multiple independent time series of length $n$ and dimension $p$, while assuming simultaneous changes in all univariate time series and an unknown number of changes. Our aim is to partition data into segments, such that in each segment the parameter associated to each time series is constant. For a time series $y$ we write $y = y_{1:n} = (y_1, \ldots, y_n) \in (\mathbb{R}^p)^n$ with $y_i^k$ the $k$-th component of the $p$-dimensional point $y_i \in \mathbb{R}^p$ in position $i$ in vector $y_{1:n}$. We also use the notation $y_{i:j} = (y_i, \ldots, y_j)$ to denote points from index $i$ to $j$. If we

assume that there are $M$ change points in a time series, this corresponds to time series splits into $M + 1$ distinct segments. The data points of each segment $m \in \{1, ..., M + 1\}$ are generated by independent random variables from a multivariate distribution with the segment-specific parameter $\theta_m = (\theta_m^1, ..., \theta_m^p) \in \mathbb{R}^p$. A segmentation with $M$ change points is defined by the vector of integers $\boldsymbol{\tau} = (\tau_0 = 0, \tau_1, ..., \tau_M, \tau_{M+1} = n)$. Segments are given by the sets of indices $\{\tau_i + 1, ..., \tau_{i+1}\}$ with $i$ in $\{0, 1, ..., M\}$.

We define the set $S_n^M$ of all possible change point locations related to the segmentation of data points between positions 1 to $n$ in $M + 1$ segments as

$$S_n^M = \{\boldsymbol{\tau} = (\tau_0, \tau_1, ..., \tau_M, \tau_{M+1}) \in \mathbb{N}^{M+2} | 0 = \tau_0 < \tau_1 < \cdots < \tau_M < \tau_{M+1} = n\}.$$

For any segmentation $\boldsymbol{\tau}$ in $S_n^M$ we define its size as $|\boldsymbol{\tau}| = M$. We denote $\mathscr{S}_n^\infty$ as the set of all possible segmentations of $y_{1:n}$:

$$\mathscr{S}_n^\infty = \bigcup_{M<n} S_n^M,$$

and take the convention that $S_n^{\infty - 1} = S_n^\infty$. In our case the number of changes $M$ is unknown, and has to be estimated.

Many approaches to detecting change points define a cost function for segmentation using the negative log-likelihood (times two). Here the negative log-likelihood (times two) calculated at the data point $y_j$ is given by function $\theta \mapsto \Omega(\theta, y_j)$, where $\theta = (\theta^1, ..., \theta^p) \in \mathbb{R}^p$. Over a segment from $i$ to $t$, the parameter remains the same and the segment cost $\mathscr{C}$ is given by

$$\mathscr{C}(y_{i:t}) = \min_{\theta \in \mathbb{R}^p} \sum_{j=i}^{t} \Omega(\theta, y_j) = \min_{\theta \in \mathbb{R}^p} \sum_{j=i}^{t} \left( \sum_{k=1}^{p} \omega(\theta^k, y_j^k) \right), \tag{1}$$

with $\omega$ the atomic likelihood function associated with $\Omega$ for each univariate time series. This decomposition is made possible by the independence hypothesis between univariate time series}. Notice that it could have been dimension-dependent with a mixture of different distributions (Gauss, Poisson, negative binomial, etc.). In our study, we use the same data model for all dimensions.

In summary, the methodology we propose relies on the assumption that:

1. the cost is point additive (see first equality in equation (1));
2. the per-point cost $\Omega$ has a simple decomposition : $\Omega(\theta) = \sum_p \omega(\theta^p)$;
3. the $\omega$ is convex.

We get that for any $\boldsymbol{\tau} \in \mathscr{S}_n^\infty$ its segmentation cost is the sum of segment cost functions:

$$\sum_{i=0}^{|\boldsymbol{\tau}|} \mathscr{C}(y_{(\tau_i+1):\tau_{i+1}}).$$

We consider a penalized version of the segment cost by a penalty $\beta > 0$, as the zero penalty case would lead to segmentation with $n$ segments. The optimal penalized cost associated with our segmentation problem is then defined by

$$\hat{Q}_n = \min_{\boldsymbol{\tau} \in S_n^\infty} \sum_{i=0}^{|\boldsymbol{\tau}|} \{\mathscr{C}(y_{(\tau_i+1):\tau_{i+1}}) + \beta\}. \tag{2}$$

The optimal segmentation $\tau$ is obtained by the argminimum in equation (2).

Various penalty forms have been proposed in the literature (Yao 1984; Killick, Fearnhead, and Eckley 2012; Zhang and Siegmund 2007; Lebarbier 2005; Verzelen et al. 2020). Summing over all segments in Equation (2), we end up with a global penalty of the form $\beta(M+1)$. Hence, our model only allows penalties that are proportional to the number of segments (Yao 1984; Killick, Fearnhead, and Eckley 2012). Penalties such as (Zhang and Siegmund 2007; Lebarbier 2005; Verzelen et al. 2020) cannot be considered with our algorithm.

By default, we set the penalty $\beta$ for $p$-variate time series of length $n$ using the Schwarz Information Criterion from (Yao 1984) (calibrated to the $p$ dimensions), as $\beta = 2p\sigma^2 \log n$. In practice, if the variance $\sigma^2$ is unknown, it is replaced by an appropriate estimation (e.g. (Hampel 1974; Hall, Kay, and Titterington 1990) as in (Lavielle and Lebarbier 2001; Liehrmann et al. 2023)).

## 1.2 Functional Pruning Optimal Partitioning Algorithm

The idea of the Optimal Partitioning (OP) method (Jackson et al. 2005) is to search for the last change point defining the last segment in data $y_{1:t}$ at each iteration (with $Q_0 = 0$), which leads to the recursion:

$$Q_t = \min_{i \in \{0,\dots,t-1\}} \left( Q_i + \mathscr{C}(y_{(i+1:t)}) + \beta \right).$$

The Pruned Exact Linear Time (PELT) method, introduced in (Killick, Fearnhead, and Eckley 2012), uses inequality-based pruning. It essentially relies on the assumption that splitting a segment in two is always beneficial in terms of cost, this is $C(y_{(i+1):j}) + C(y_{(j+1):t}) \le C(y_{(i+1):t})$. This assumption is always true in our setting. PELT considers each change point candidate sequentially and decides whether $i$ can be excluded from the set of changepoint candidates if $\hat{Q}_i + \mathscr{C}(y_{(i+1):t}) \ge \hat{Q}_t$, as $i$ cannot appear as the optimal change point in future iterations.

*Functional description.* In the FPOP method we introduce a last segment parameter $\theta = (\theta^1, \dots, \theta^p)$ in $\mathbb{R}^p$ and define a functional cost $\theta \mapsto Q_t(\theta)$ depending on $\theta$, that takes the following form:

$$Q_t(\theta) = \min_{\tau \in S_t} \left( \sum_{i=0}^{M-1} \{ \mathscr{C}(y_{(\tau_i+1):\tau_{i+1}}) + \beta \} + \sum_{j=\tau_M+1}^{t} \Omega(\theta, y_j) + \beta \right).$$

As explained in (Maidstone et al. 2017), we can compute the function $Q_{t+1}(\cdot)$ based only on the knowledge of $Q_t(\cdot)$ for each integer $t$ from 0 to $n-1$. We have:

$$Q_{t+1}(\theta) = \min\{Q_t(\theta), \hat{Q}_t + \beta\} + \Omega(\theta, y_{t+1}), \tag{3}$$

for all $\theta \in \mathbb{R}^p$, with $\hat{Q}_t = \min_\theta Q_t(\theta)$ ($t \ge 1$) and the initialization $Q_0(\theta) = \beta$, $\hat{Q}_0 = 0$ so that $Q_1(\theta) = \Omega(\theta, y_1) + \beta$. By looking closely at this relation, we see that each function $Q_t$ is a piece-wise continuous function consisting of at most $t$ different functions on $\mathbb{R}^p$, denoted $q_t^i$:

$$Q_t(\theta) = \min_{i \in \{1,\dots,t\}} \{ q_t^i(\theta) \},$$

where the $q_t^i$ functions are given by explicit formulas:

$$q_t^i(\theta) = \hat{Q}_{i-1} + \beta + \sum_{j=i}^{t} \Omega(\theta, y_j), \quad \theta \in \mathbb{R}^p, \quad i = 1, \dots, t.$$

and

$$\hat{Q}_{i-1} = \min_{\theta \in \mathbb{R}^p} Q_{i-1}(\theta) = \min_{j \in \{1, \dots, i-1\}} \left\{ \min_{\theta \in \mathbb{R}^p} q_{i-1}^j(\theta) \right\}. \tag{4}$$

It is important to notice that each $q_t^i$ function is associated with the last change point $i-1$ and the last segment is given by indices from $i$ to $t$. Consequently, the last change point at step $t$ in $y_{1:t}$ is denoted as $\hat{\tau}_t$ ($\hat{\tau}_t \leq t-1$) and is given by

$$\hat{\tau}_t = Arg \min_{i \in \{1, \dots, t\}} \left\{ \min_{\theta \in \mathbb{R}^p} q_t^i(\theta) \right\} - 1.$$

*Backtracking.* Knowing the values of $\hat{\tau}_t$ for all $t = 1, \dots, n$, we can always restore the optimal segmentation at time $n$ for $y_{1:n}$. This procedure is called backtracking. The vector $cp(n)$ of ordered change points in the optimal segmentation of $y_{1:n}$ is determined recursively by the relation $cp(n) = (cp(\hat{\tau}_n), \hat{\tau}_n)$ with stopping rule $cp(0) = \emptyset$.

*Parameter space description.* Applying functional pruning requires a precise analysis of the recursion (3) that depends on the property of the cost function $\Omega$. In what follows we consider three choices based on a Gaussian, Poisson, and negative binomial distribution for data distribution. The exact formulas of these cost functions are given in Section 5.1.

We denote the set of parameter values for which the function $q_t^i(\cdot)$ is optimal as:

$$Z_t^i = \left\{ \theta \in \mathbb{R}^p | Q_t(\theta) = q_t^i(\theta) \right\}, \quad i = 1, \dots, t.$$

also called the *living zone.* The key idea behind functional pruning is that the $Z_t^i$ are nested ($Z_{t+1}^i \subset Z_t^i$) thus as soon as we can prove the emptiness of one set $Z_t^i$, we delete its associated $q_t^i$ function and do not have to consider its minimum anymore at any further iteration (proof in Section 1.3). In dimension $p = 1$ this is reasonably easy. In this case, the sets $Z_t^i$ ($i = 1, \dots, t$) are unions of intervals and an efficient functional pruning rule is possible by updating a list of these intervals for $Q_t$. This approach is implemented in FPOP (Maidstone et al. 2017).

In dimension $p \geq 2$ it is not so easy anymore to keep track of the emptiness of the sets $Z_t^i$. We illustrate the dynamics of the $Z_t^i$ sets in Figure 1 in the bivariate Gaussian case. Each color is associated with a set $Z_t^i$ (corresponding to a possible change at $i-1$) for $t$ equal 1 to 5. This plot shows in particular that sets $Z_t^i$ can be non-convex.

## 1.3 Geometric Formulation of Functional Pruning

To build an efficient pruning strategy for dimension $p \geq 2$ we need to test the emptiness of the sets $Z_t^i$ at each iteration. Note that to get $Z_t^i$ we need to compare the functional cost $q_t^i$ with any other functional cost $q_t^j$, $j = 1, \dots, t$, $j \neq i$. This leads to the definition of the following sets.

**Definition 1.1.** We define *S-type set* $S_j^i$ using the function $\Omega$ as

$$S_j^i = \left\{ \theta \in \mathbb{R}^p \mid \sum_{u=i}^{j-1} \Omega(\theta, y_u) \leq \hat{Q}_{j-1} - \hat{Q}_{i-1} \right\}, \quad \text{when } i < j$$

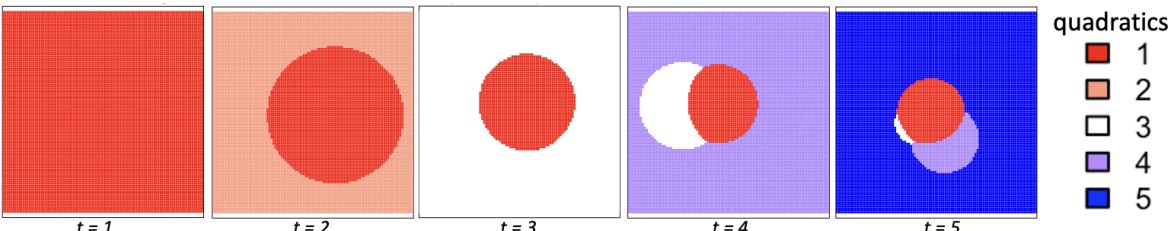

Figure 1: The sets $Z_t^i$ over time for the bivariate independent Gaussian model on time series without change $y = ((0.29, 1.93), (1.86, -0.02), (0.9, 2.51), (-1.26, 0.91), (1.22, 1.11))$. From left to right we represent at time $t = 1, 2, 3, 4$, and 5 the parameter space $(\theta^1, \theta^2)$. Each $Z_t^i$ is represented by a color. The change 1 associated with quadratics 2 is pruned at $t = 3$. Notice that each time sequence of $Z_t^i$ with $i$ fixed is a nested sequence of sets.

and $S_i^i = \mathbb{R}^p$. We denote the set of all possible S-type sets as **S**.

To ease some of our calculations, we now introduce some additional notations. For $\theta = (\theta^1, \ldots, \theta^p)$ in $\mathbb{R}^p$, $1 \le i < j \le n$ we define $p$ univariate functions $\theta^k \mapsto s_{ij}^k(\theta^k)$ associated to the $k$-th time series as

$$s_{ij}^k(\theta^k) = \sum_{u=i}^{j-1} \omega(\theta^k, y_u^k), \quad k = 1, \ldots, p. \tag{5}$$

We introduce a constant $\Delta_{ij}$ and a function $\theta \mapsto s_{ij}(\theta)$:

$$\begin{cases} \Delta_{ij} = \hat{Q}_{j-1} - \hat{Q}_{i-1}, \\ s_{ij}(\theta) = \sum_{k=1}^{p} s_{ij}^k(\theta^k) - \Delta_{ij}, \end{cases} \tag{6}$$

where $\hat{Q}_{i-1}$ and $\hat{Q}_{j-1}$ are defined as in (4). The sets $S_j^i$ for $i < j$ can thus be written as

$$S_j^i = s_{ij}^{-1}(-\infty, 0]. \tag{7}$$

In Figure 2 we present the level curves for three different parametric models given by $s_{ij}^{-1}(\{w\})$ with $w$ a real number. Each of these curves encloses an S-type set, which, according to the definition of the function $\omega$, is convex.

At time $t = 1, \ldots, n$ we define the following sets associated to the last change point index $i - 1$:

-past set $\mathcal{P}^i$

$$\mathcal{P}^i = \{S_i^u, u = 1, \ldots, i - 1\}.$$

-future set $\mathcal{F}^i(t)$

$$\mathcal{F}^i(t) = \{S_v^i, v = i, \ldots, t\}.$$

We denote the cardinal of a set $\mathcal{A}$ as $|\mathcal{A}|$. Using these two sets of sets, the $Z_t^i$ have the following description.

**Proposition 1.1.** *At iteration $t$, the living zones $Z_t^i$ ($i = 1, \ldots, t$) are defined by the functional cost $Q_t(\cdot)$, with each of them being formed as the intersection of sets in $\mathcal{F}^i(t)$ excluding the union of sets in $\mathcal{P}^i$.*

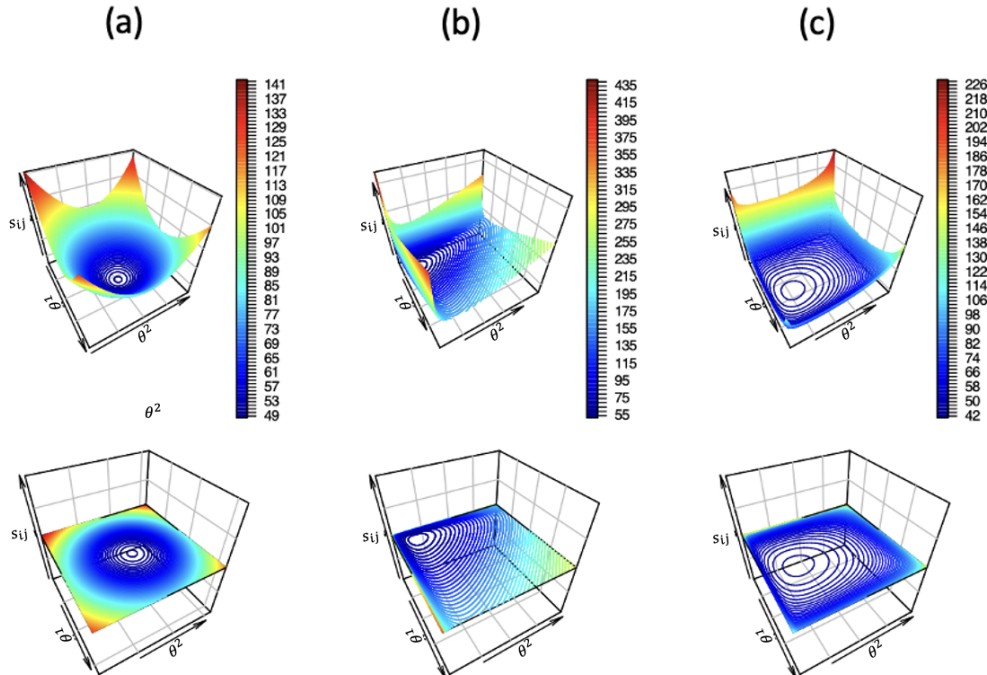

Figure 2: Three examples of the level curves of a function $s_{ij}$ for bivariate time series $\{y^1, y^2\}$. We use the following simulations for univariate time series : (a) $y^1 \sim \mathcal{N}(0, 1)$, $y^2 \sim \mathcal{N}(0, 1)$, (b) $y^1 \sim \mathcal{P}(1)$, $y^2 \sim \mathcal{P}(3)$, (c) $y^1 \sim \mathcal{NB}(0.5, 1)$, $y^2 \sim \mathcal{NB}(0.8, 1)$.

$$Z_t^i = \Big( \bigcap_{S \in \mathcal{F}^i(t)} S \Big) \setminus \big( \cup_{S \in \mathcal{P}^i} S \big), \quad i = 1, \ldots, t. \tag{8}$$

*Proof.* Based on the definition of the set $Z_t^i$, the proof is straightforward. Parameter value $\theta$ is in $Z_t^i$ if and only if $q_t^i(\theta) \leq q_t^u(\theta)$ for all $u \neq i$; these inequalities define the past set (when $u < i$) and the future set (when $u \geq i$).

$\square$

Proposition 1.1 states that regardless of the value of i, the living zone $Z_t^i$ is formed through intersection and elimination operations on $t$ S-type sets. Notably, one of these sets, $S_i^i$, always represents the entire space $\mathbb{R}^p$.

**Corollary 1.1.** *The sequence $\zeta^i = (Z_t^i)_{t \geq i}$ is a nested sequence of sets.*

Indeed, $Z_{t+1}^i$ is equal to $Z_t^i$ with an additional intersection in the future set. Based on Corollary 1.1, as soon as we prove that the set $Z_t^i$, is empty, we delete its associated $q_t^i$ function and, consequently, we can prune the change point $i - 1$. In this context, functional and inequality-based pruning have a simple geometric interpretation.

*Functional pruning geometry.* The position $i - 1$ is pruned at step $t$, in $Q_t(\cdot)$, if the intersection set of $\bigcap_{S \in \mathcal{F}^i(t)} S$ is covered by the union set $\cup_{S \in \mathcal{P}^i} S$.

*Inequality-based pruning geometry.* The inequality-based pruning of PELT is equivalent to the geometric rule: position $i - 1$ is pruned at step $t$ if the set $S_t^i$ is empty. In that case, the intersection set $\bigcap_{S \in \mathcal{F}^i(t)} S$ is empty, and therefore $Z_t^i$ is also empty using (8). This shows that if a change is pruned

using inequality-based pruning it is also pruned using functional pruning. For the dimension $p = 1$ this claim was theoretically proved in (Maidstone et al. 2017).

According to Proposition 1.1, beginning with $Z_i^i = \mathbb{R}^p$, the set $Z_t^i$ is derived by iteratively applying two types of operations: intersection with an S-type set $S$ from $\mathscr{F}^i(t)$ or subtraction of an S-type set $S$ from $\mathscr{P}^i$. The construction of set $Z_t^i$ using Proposition 1.1 is illustrated in Figure 3 for a bivariate independent Gaussian case: we have the intersection of three S-type sets and the subtraction of three S-type sets. This simple example highlights that the set $Z_t^i$ is typically non-convex, posing challenge in studying its emptiness.

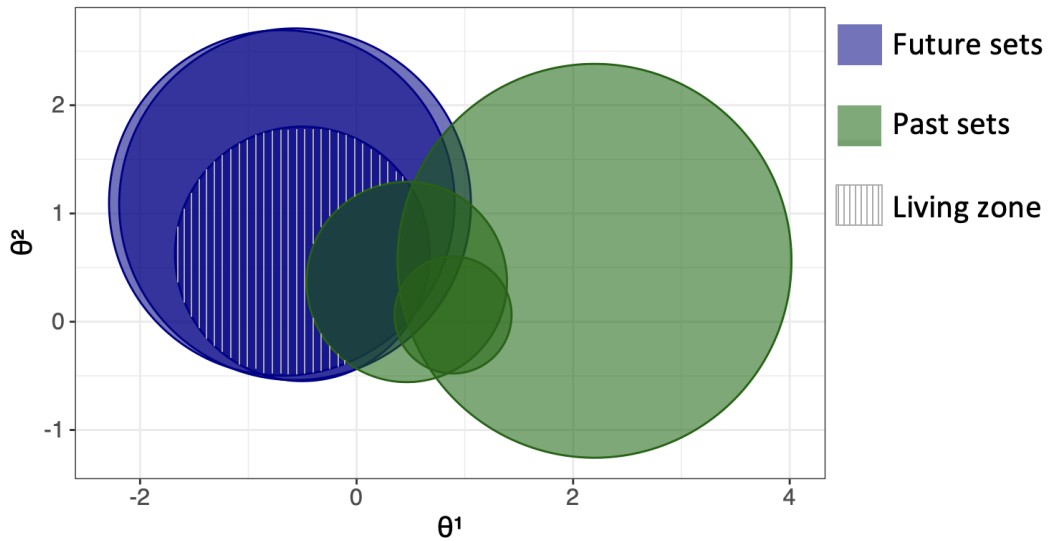

Figure 3: Examples of building a living zone $Z_t^i$ with $|\mathscr{P}^i| = |\mathscr{F}^i(t)| = 3$ for the Gaussian case in 2-D ($\mu = 0, \sigma = 1$). The green disks are S-type sets of the past set $\mathscr{P}^i$. The blue disks are S-type sets of the future set $\mathscr{F}^i(t)$. The shaded area is the set $Z_t^i$.

## 2 Geometric Functional Pruning Optimal Partitioning

### 2.1 General Principle of GeomFPOP

Rather than considering an exact representation of the $Z_t^i$, our idea is to consider a hopefully slightly larger set that is easier to update. To be specific, for each $Z_t^i$ we introduce $\tilde{Z}_t^i$, called *testing set*, such that $Z_t^i \subset \tilde{Z}_t^i$. If at time $t$ $\tilde{Z}_t^i$ is empty thus is $Z_t^i$ and thus change $i - 1$ can be pruned. From Proposition 1.1 we have that starting from $Z_i^i = \mathbb{R}^p$ the set $Z_t^i$ is obtained by successively applying two types of operations: intersection with an S-type set $S$ ($Z \bigcap S$) or subtraction of an S-type set $S$ ($Z \setminus S$). Similarly, starting from $\tilde{Z}_i^i = \mathbb{R}^p$ we obtain $\tilde{Z}_t^i$ by successively applying approximation of these intersection and subtraction operations. Intuitively, the complexity of the resulting algorithm is a combination of the efficiency of the pruning and the easiness of updating the testing set.

*A Generic Formulation of GeomFPOP.* In what follows we will generically describe GeomFPOP, that is, without specifying the precise structure of the testing set $\tilde{Z}_t^i$. We call $\widetilde{\mathbf{Z}}$ the set of all possible $\tilde{Z}_t^i$ and assume the existence of two operators $\bigcap_{\tilde{Z}}$ and $\setminus_{\tilde{Z}}$. We have the following assumptions for these operators.

**Definition 2.1.** The two operators $\bigcap_{\tilde{Z}}$ and $\setminus_{\tilde{Z}}$ are such that:

1. the left input is a $\tilde{Z}$-type set (that is an element of $\widetilde{\mathbf{Z}}$);
2. the right input is a $S$-type set;

3. the output is a $\tilde{Z}$-type set;
4. $\tilde{Z} \bigcap S \subset \tilde{Z} \bigcap_{\tilde{Z}} S$ and $\tilde{Z} \setminus S \subset \tilde{Z} \setminus_{\tilde{Z}} S$.

We give a proper description of two types of testing sets and their approximation operators in Section 3.

At each iteration $t$ GeomFPOP will construct $\tilde{Z}_t^i$ (with $i < t$) from $\tilde{Z}_{t-1}^i$, $\mathscr{P}^i$ and $\mathscr{F}^i(t)$ iteratively using the two operators $\bigcap_{\tilde{Z}}$ and $\setminus_{\tilde{Z}}$. To be specific, we define $S_j^F$ the j-th element of $\mathscr{F}^i(t)$ and $S_P^j$ the j-th element of $\mathscr{P}^i$, we use the following iterations:

$$
\begin{cases}
A_0 = \tilde{Z}_t^i, \quad A_j = A_{j-1} \bigcap_{\tilde{Z}} S_j^F, \quad j = 1, \dots, |\mathscr{F}^i(t)|, \\
B_0 = A_{|\mathscr{F}^i(t)|}, \quad B_j = B_{j-1} \setminus_{\tilde{Z}} S_P^j, \qquad j = 1, \dots, |\mathscr{P}^i|,
\end{cases}
$$

and define $\tilde{Z}_t^i = B_{|\mathscr{P}^i|}$. Using the fourth property of Definition 2.1 and Proposition 1.1, we get that at any time of the algorithm $\tilde{Z}_t^i$ contains $Z_t^i$.

The pseudo-code of this procedure is described in Algorithm 1. The select($\mathscr{A}$) step in Algorithm 1, where $\mathscr{A} \subset \mathbf{S}$, returns a subset of $\mathscr{A}$ in $\mathbf{S}$. By default, select($\mathscr{A}$) := $\mathscr{A}$.

---

**Algorithm 1** Geometric update rule of $\tilde{Z}_t^i$

---

   **procedure** UPDATEZONE($\tilde{Z}_{t-1}^i, \mathscr{P}^i, \mathscr{F}^i(t), i < t$)
      $\tilde{Z}_t^i \leftarrow \tilde{Z}_{t-1}^i$
      **for** $S \in$ select($\mathscr{F}^i(t-1)$) **do**
         $\tilde{Z}_t^i \leftarrow \tilde{Z}_t^i \bigcap_{\tilde{Z}} S$
      **for** $S \in$ select($\mathscr{P}^i$) **do**
         $\tilde{Z}_t^i \leftarrow \tilde{Z}_t^i \setminus_{\tilde{Z}} S$
      **return** $\tilde{Z}_t^i$

---

We denote the set of candidate change points at time $t$ as $\tau_t$. Note that for any $(i-1) \in \tau_t$ the sum of $|\mathscr{P}^i|$ and $|\mathscr{F}^i(t)|$ is $|\tau_t|$. With the default select procedure we do $\mathcal{O}(p|\tau_t|)$ operations in the updateZone procedure. By limiting the number of elements returned by select we can reduce the complexity of the updateZone procedure.

*Remark.* For example, if the operator $\mathscr{A} \mapsto$ select($\mathscr{A}$), regardless of $|\mathscr{A}|$, always returns a subset of constant size, then the overall complexity of GeomFPOP is at worst $\sum_{t=1}^{n} \mathcal{O}(p|\tau_t|)$.

Using this updateZone procedure we can now informally describe the GeomFPOP algorithm. At each iteration the algorithm will

1. find the minimum value for $Q_t$, $m_t$ and the best position for last change point $\hat{\tau}_t$ (note that this step is standard: as in the PELT algorithm we need to minimize the cost of the last segment defined in equation (1));
2. compute all sets $\tilde{Z}_t^i$ using $\tilde{Z}_{t-1}^i$, $\mathscr{P}^i$, and $\mathscr{F}^i(t)$ with the updateZone procedure;
3. remove changes such that $\tilde{Z}_t^i$ is empty.

To simplify the pseudo-code of GeomFPOP, we also define the following operators:

1. bestCost&Tau($t$) operator returns two values: the minimum value of $Q_t$, $m_t$, and the best position for last change point $\hat{\tau}_t$ at time $t$ (see Section 1.2);
2. getPastFutureSets($i, t$) operator returns a pair of sets $(\mathscr{P}^i, \mathscr{F}^i(t))$ for change point candidate $i - 1$ at time $t$;
3. backtracking($\hat{\tau}, n$) operator returns the optimal segmentation for $y_{1:n}$.

The pseudo-code of GeomFPOP is presented in Algorithm 2.

---

**Algorithm 2** GeomFPOP algorithm

---

**procedure** GEOMFPOP($y, \Omega(\cdot, \cdot), \beta$)

    $\hat{Q}_0 \leftarrow 0, \quad Q_0(\theta) \leftarrow \beta, \quad \tau_0 \leftarrow \emptyset, \quad \{\tilde{Z}_i^i\}_{i \in \{1,\dots,n\}} \leftarrow \mathbb{R}^p$

    **for** $t = 1, \dots, n$ **do**

        $Q_t(\theta) \leftarrow \min\{Q_{t-1}(\theta), \hat{Q}_{t-1} + \beta\} + \Omega(\theta, y_t)$

        $(\hat{Q}_t, \hat{\tau}_t) \leftarrow \texttt{bestCost\&Tau}(t)$

        **for** $i - 1 \in \tau_{t-1}$ **do**

            $(\mathscr{P}^i, \mathscr{F}^i(t)) \leftarrow \texttt{getPastFutureSets}(i, t)$

            $\tilde{Z}_t^i \leftarrow \texttt{updateZone}(\tilde{Z}_{t-1}^i, \mathscr{P}^i, \mathscr{F}^i(t), i, t)$

            **if** $\tilde{Z}_t^i = \emptyset$ **then**

                $\tau_{t-1} \leftarrow \tau_{t-1} \backslash \{i - 1\}$

        $\tau_t \leftarrow (\tau_{t-1}, t - 1)$

    **return** $cp(n) \leftarrow \texttt{backtracking}(\hat{\tau} = (\hat{\tau}_1, \dots, \hat{\tau}_n), n)$

---

*Remark.* Whatever the number of elements returned by the `select` operator for computing $\tilde{Z}_t^i$, we can guarantee the exactness of the GeomFPOP algorithm, since the approximate living zone (the testing set) includes the living zone (8), as we consider less intersections and set subtractions.

# 3 Approximation Operators $\bigcap_{\tilde{Z}}$ and $\setminus_{\tilde{Z}}$

The choice of the geometric structure and the way it is constructed directly affects the computational cost of the algorithm. We consider two types of testing set $\tilde{Z} \in \widetilde{\mathbf{Z}}$, a S-type set $\tilde{S} \in \mathbf{S}$ (see Definition 1.1) and a hyperrectangle $\tilde{R} \in \mathbf{R}$ defined below.

**Definition 3.1.** Given two vectors in $\mathbb{R}^p$, $\tilde{l}$ and $\tilde{r}$ we define the set $\tilde{R}$, called *hyperrectangle*, as:

$$\tilde{R} = [\tilde{l}_1, \tilde{r}_1] \times \cdots \times [\tilde{l}_p, \tilde{r}_p] \,.$$

We denote the set of all possible sets $\tilde{R}$ as $\mathbf{R}$.

To update the testing sets we need to give a strict definition of the operators $\bigcap_{\tilde{Z}}$ and $\setminus_{\tilde{Z}}$ for each type of testing set. To facilitate the following discussion, we rename them. For the first type of geometric structure, we rename the testing set $\tilde{Z}$ as $\tilde{S}$, the operators $\bigcap_{\tilde{Z}}$ and $\setminus_{\tilde{Z}}$ as $\bigcap_S$ and $\setminus_S$ and $\tilde{Z}$-type approximation as S-type approximation. And, likewise, we rename the testing set $\tilde{Z}$ as $\tilde{R}$, the operators $\bigcap_{\tilde{Z}}$ and $\setminus_{\tilde{Z}}$ as $\bigcap_R$ and $\setminus_R$ and $\tilde{Z}$-type approximation as R-type approximation for the second type of geometric structure.

## 3.1 S-type Approximation

With this approach, our goal is to keep track of the fact that at time $t = 1, \dots, n$ there is a pair of changes $(u_1, u_2)$, with $u_1 < i < u_2 \leq t$ such that $S_{u_2}^i \subset S_i^{u_1}$ or there is a pair of changes $(v_1, v_2)$, with $i < v_1 < v_2 \leq t$ such that $S_{v_1}^i \bigcap S_{v_2}^i$ is empty. If at time $t$ at least one of these conditions is met, we can guarantee that the set $\tilde{S}$ is empty, otherwise, we propose to keep as the result of approximation the last future S-type set $S_t^i$, because it always includes the set $Z_t^i$. This allows us to quickly check and prove (if $\tilde{S} = \emptyset$) the emptiness of set $Z_t^i$.

We consider two generic S-type sets, $S$ and $\tilde{S}$ from $\mathbf{S}$, described as in Definition 1.1 by the functions $s$ and $\tilde{s}$:

$$s(\theta) = \sum_{k=1}^{p} s^k(\theta^k) - \Delta, \qquad \tilde{s}(\theta) = \sum_{k=1}^{p} \tilde{s}^k(\theta^k) - \tilde{\Delta}.$$

**Definition 3.2.** For all $S$ and $\tilde{S}$ in **S** we define the operators $\bigcap_S$ and $\setminus_S$ as:

$$\tilde{S} \bigcap_S S = \begin{cases} \varnothing, & \text{if } \tilde{S} \bigcap S = \varnothing, \\ \tilde{S}, & \text{otherwise}. \end{cases}$$

$$\tilde{S} \setminus_S S = \begin{cases} \varnothing, & \text{if } \tilde{S} \subset S, \\ \tilde{S}, & \text{otherwise}. \end{cases}$$

As a consequence, we only need an easy way to detect any of these two geometric configurations: $\tilde{S} \bigcap S$ and $\tilde{S} \subset S$.

In the Gaussian case, the S-type sets are $p$-balls and an easy solution exists based on comparing radii (see Section 5.2 for details). In the case of other models (as Poisson or negative binomial), intersection and inclusion tests can be performed based on a solution using separative hyperplanes and iterative algorithms for convex problems (see Section 5.3). We propose another type of testing set solving all types of models with the same method.

## 3.2 R-type Approximation

Here, we approximate the sets $Z_t^i$ by hyperrectangles $\tilde{R}_t^i \in \mathbf{R}$. A key insight of this approximation is that given a hyperrectangle $R$ and an S-type set $S$ we can efficiently (in $\mathcal{O}(p)$ using Proposition 3.2) recover the best hyperrectangle approximation of $R \cup S$ and $R \setminus S$. Formally we define these operators as follows.

**Definition 3.3.** For all $R, \tilde{R} \in \mathbf{R}$ and $S \in \mathbf{S}$ we define the operators $\bigcap_R$ and $\setminus_R$ as:

$$R \bigcap_R S = \bigcap_{\{\tilde{R} \mid R \bigcap S \subset \mathbf{R}\}} \tilde{R},$$

$$R \setminus_R S = \bigcap_{\{\tilde{R} \mid R \setminus S \subset \mathbf{R}\}} \tilde{R}.$$

We now explain how we compute these two operators. First, we note that they can be recovered by solving $2p$ one-dimensional optimization problems.

**Proposition 3.1.** *The $k$-th minimum coordinates $\tilde{l}_k$ and maximum coordinates $\tilde{r}_k$ of $\tilde{R} = R \bigcap_R S$ (resp. $\tilde{R} = R \setminus_R S$) is obtained as*

$$\tilde{l}_k \text{ or } \tilde{r}_k = \begin{cases} \min_{\theta_k \in \mathbb{R}} \text{ or } \max_{\theta_k \in \mathbb{R}} \theta_k, \\ \text{subject to } \varepsilon s(\theta) \leq 0, \\ \qquad\qquad l_j \leq \theta_j \leq r_j, \quad j = 1, \dots, p, \end{cases} \tag{9}$$

*with $\varepsilon = 1$ (resp. $\varepsilon = -1$).*

To solve the previous problems ($\varepsilon = 1$ or $-1$), we define the following characteristic points.

**Definition 3.4.** Let $S \in \mathbf{S}$, described by function $s(\theta) = \sum_{k=1}^{p} s^k(\theta^k) - \Delta$ from the family of functions (6), with $\theta \in \mathbb{R}^p$. We define the *minimal point* $\mathbf{c} \in \mathbb{R}^p$ of $S$ as:

$$\mathbf{c} = \left\{\mathbf{c}^k\right\}_{k=1,\dots,p}, \qquad \text{with} \qquad \mathbf{c}^k = \underset{\theta^k \in \mathbb{R}}{Arg\min}\{s^k(\theta^k)\}. \tag{10}$$

Moreover, with $R \in \mathbf{R}$ defined through vectors $l, r \in \mathbb{R}^p$, we define two points of $R$, the *closest point* $\mathbf{m} \in \mathbb{R}^p$ and the *farthest point* $\mathbf{M} \in \mathbb{R}^p$ relative to $S$ as

$$\mathbf{m} = \left\{\mathbf{m}^k\right\}_{k=1,\dots,p}, \qquad \text{with} \qquad \mathbf{m}^k = \underset{l^k \le \theta^k \le r^k}{Arg\min}\left\{s^k(\theta^k)\right\},$$

$$\mathbf{M} = \left\{\mathbf{M}^k\right\}_{k=1,\dots,p}, \qquad \text{with} \qquad \mathbf{M}^k = \underset{l^k \le \theta^k \le r^k}{Arg\max}\left\{s^k(\theta^k)\right\}.$$

*Remark.* In the Gaussian case, $S$ is a ball in $\mathbb{R}^p$ and

- $\mathbf{c}$ is the center of the ball;
- $\mathbf{m}$ is the closest point to $\mathbf{c}$ inside $R$;
- $\mathbf{M}$ is the farthest point to $\mathbf{c}$ in $R$.

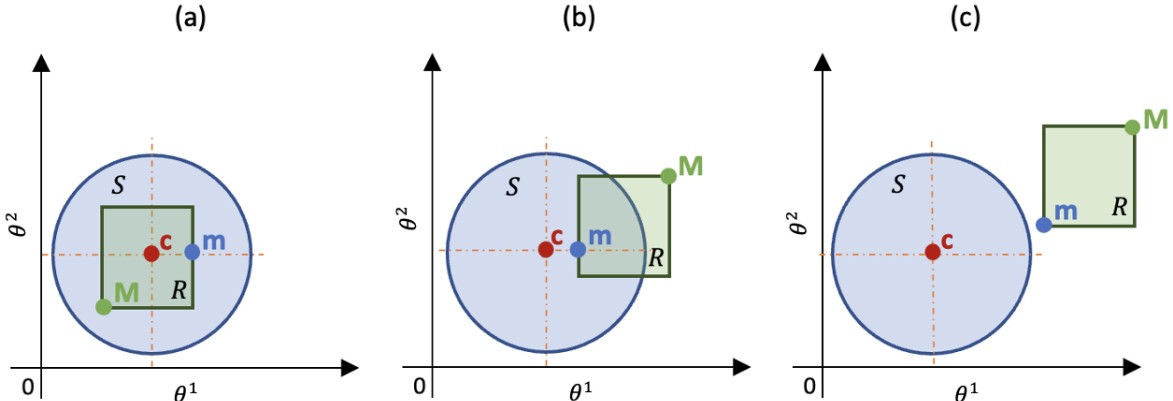

Figure 4: Three examples of minimal point $\mathbf{c}$, closest point $\mathbf{m}$ and farthest point $\mathbf{M}$ for bivariate Gaussian case: (a) $R \subset S$; (b) $R \bigcap S \ne \varnothing$; (c) $R \bigcap S = \varnothing$.

**Proposition 3.2.** *Let $\tilde{R} = R \bigcap_R S$ (resp. $R \setminus_R S$), with $R \in \mathbf{R}$ and $S \in \mathbf{S}$. We compute the boundaries $(\tilde{l}, \tilde{r})$ of $\tilde{R}$ using the following rule:*

1. *We define the point $\tilde{\theta} \in \mathbb{R}^p$ as the closest point $\mathbf{m}$ (resp. farthest $\mathbf{M}$). For all $k = 1, \dots p$ we find the roots $\theta^{k_1}$ and $\theta^{k_2}$ of the one-variable ($\theta^k$) equation*

$$s^k(\theta^k) + \sum_{j \ne k} s^j(\tilde{\theta}^j) - \Delta = 0.$$

*If the roots are real-valued we consider that $\theta^{k_1} \le \theta^{k_2}$, otherwise we write $\left[\theta^{k_1}, \theta^{k_2}\right] = \varnothing$.*

2. *We compute the boundary values $\tilde{l}^k$ and $\tilde{r}^k$ of $\tilde{R}$ as:*

- *For $R \bigcap_R S$ ($k = 1, \dots, p$):*

$$\left[\tilde{l}^k, \tilde{r}^k\right] = \left[\theta^{k_1}, \theta^{k_2}\right] \bigcap \left[l^k, r^k\right]. \tag{11}$$

- *For $R \setminus_R S$ ($k = 1, \ldots, p$):*

$$\left[\tilde{l}^k, \tilde{r}^k\right] = \begin{cases} \left[l^k, r^k\right] \setminus \left[\theta^{k_1}, \theta^{k_2}\right], & if \quad \left[\theta^{k_1}, \theta^{k_2}\right] \not\subset \left[l^k, r^k\right], \\ \left[l^k, r^k\right], & otherwise. \end{cases}$$

*If there is a dimension k for which $\left[\tilde{l}^k, \tilde{r}^k\right] = \emptyset$, then the set $\tilde{R}$ is empty.*

The proof of Proposition 3.2 is presented in Section 5.4.

As a partial conclusion to this theoretical study, those ideas could be extended to some other models with missing values or dependencies between dimensions (e.g. piece-wise constant regression). However, it would require introducing new approximation operators of potential high complexity.

# 4    Simulation Study of GeomFPOP

In this section, we study the efficiency of GeomFPOP using simulations of multivariate independent time series. For this, we implemented GeomFPOP (with S and R types) and PELT for the Multivariate Independent Gaussian Model in the R-package 'GeomFPOP' https://github.com/lpishchagina/Geom FPOP written in R/C++. By default, the value of penalty $\beta$ for each simulation was defined by the Schwarz Information Criterion proposed in (Yao 1984) as $\beta = 2p\sigma^2 \log n$ with $\sigma = 1$ known. As long as the per-dimension variance is known (or appropriately estimated) we can make this assumption ($\sigma = 1$ known) without loss of generality by rescaling the data by the standard deviation.

*Overview of our simulations.* First, for $2 \leq p \leq 10$ we generated $p$-variate independent time series (multivariate independent Gaussian model with fixed variance) with $n = 10^4$ data points and number of segments: $1, 5, 10, 50$ and $100$. The segment-specific parameter (mean) was set to 1 for even segments, and 0 for odd segments. As a quality control measure, we verified that PELT and GeomFPOP produced identical outputs on these simulated profiles. Second, we studied cases where the PELT approach is not efficient, that is when the data has no or few changes relative to $n$. Indeed, it was shown in (Killick, Fearnhead, and Eckley 2012) and (Maidstone et al. 2017) that the run time of PELT is close to $\mathcal{O}(n^2)$ in such cases. So we considered simulations of multivariate time series without change (only one segment). By these simulations we evaluated the pruning efficiency of GeomFPOP (using S and R types) for dimension $2 \leq p \leq 10$ (see Figure 5 in Section 4.1). For small dimensions we also evaluated the run time of GeomFPOP and PELT and compare them (see Figure 6 in Section 4.2). In addition, we considered another approximation of the $Z_t^i$ where we applied our $\bigcap_R$ and $\setminus_R$ operators only for a randomly selected subset of the past and future balls. In practice, this strategy turned out to be faster computationally than the full/original GeomFPOP and PELT (see Figure 7 in Section 4.3). For this strategy we also generated time series of a fixed size ($10^6$ data points) and varying number of segments and evaluated how the run time vary with the number of segments for small dimensions ($2 \leq p \leq 4$). Our empirical results confirmed that the GeomFPOP (R-type: `random/random`) approach is computationally comparable to PELT when the number of changes is large (see Figure 9 in Section 4.5).

## 4.1    The Number of Change Point Candidates stored over Time

We evaluate the functional pruning efficiency of the GeomFPOP method using $p$-variate independent Gaussian noise of length $n = 10^4$ data points. For such series, PELT typically does not pruned (e.g. for $t = 10^4$, $p = 2$ it stores almost always $t$ candidates).

We report in Figure 5 the percentage of candidates that are kept by GeomFPOP as a function of $n$, $p$ and the type of pruning (R or S). Regardless of the type of approximation and contrary to PELT, we

observe that there is some pruning. However when increasing the dimension $p$, the quality of the pruning decreases.

Comparing the left plot of Figure 5 with the right plot we see that for dimensions $p = 2$ to $p = 5$ R-type prunes more than the S-type, while for larger dimensions the S-type prunes more than the R-type. For example, for $p = 2$ at time $t = 10^4$ by GeomFPOP (R-type) the number of candidates stored over $t$ does not exceed 1% versus 3% by GeomFPOP (S-type). This intuitively makes sense. One the one hand, the R-type approximation of a sphere deteriorates as the dimension increases. On the other hand with R-type approximation every new approximation is included in the previous one. For small dimensions this memory effect outweighs the roughness of the approximation.

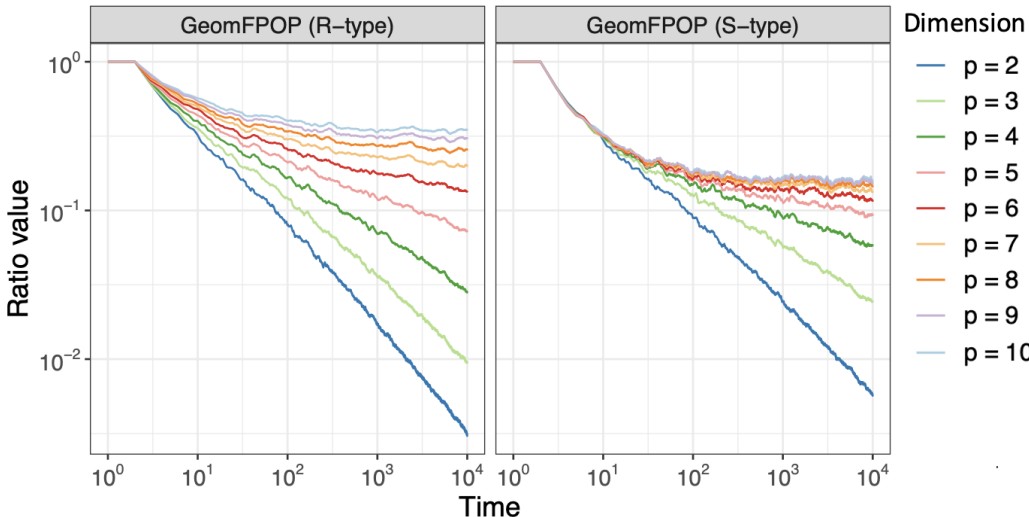

Figure 5: Percentage of candidate change points stored over time by GeomFPOP with R (left) or S (right) type pruning for dimension $p = 2, \ldots, 10$. Averaged over 100 data sets.

Based on these results we expect that R-type pruning GeomFPOP will be more efficient than S-type pruning for small dimensions.

## 4.2 Empirical Time Complexity of GeomFPOP

We studied the run time of GeomFPOP (S and R-type) and compared it to PELT for small dimensions. We simulated data generated by a $p$-variate i.i.d. Gaussian noise and saved their run times with a three minutes limit. The results are presented in Figure 6. We observe that GeomFPOP is faster than PELT only for $p = 2$. For $p = 3$ run times are comparable and for $p = 4$ GeomFPOP is slower. This is not in line with the fact that GeomFPOP prunes more than PELT. However, as explained in Section 2.1, the computational complexity of GeomFPOP and PELT is affected by both the efficiency of pruning and the number of comparisons conducted at each step. For PELT at time $t$, all candidates are compared to the last change, resulting in a complexity of order $\mathcal{O}(p|\tau_t^{PELT}|)$. On the other hand, GeomFPOP compares all candidates to each other (refer to Algorithm 1 and the remark from Section 2.1), leading to a complexity of order $\mathcal{O}(p|\tau_t^{GeomFPOP}|^2)$. In essence, the complexity of GeomFPOP is governed by the square of the number of candidates. Therefore, GeomFPOP is expected to be more efficient than PELT only if its square number of candidates is smaller than the number of candidates for PELT. Based on the information presented in} Figure 6, we argue that this condition holds true only for dimensions $p = 2$ and 3. Indeed, analysis of the number of comparisons between PELT and GeomFPOP (see Section 5.6) supports this claim, revealing that GeomFPOP (S-type) outperforms PELT only when $p \leq 2$ and GeomFPOP (R-type) outperforms PELT only when $p \leq 3$} (see Figure 12 in Section 5.6). This leads us to consider a randomized version of GeomFPOP.

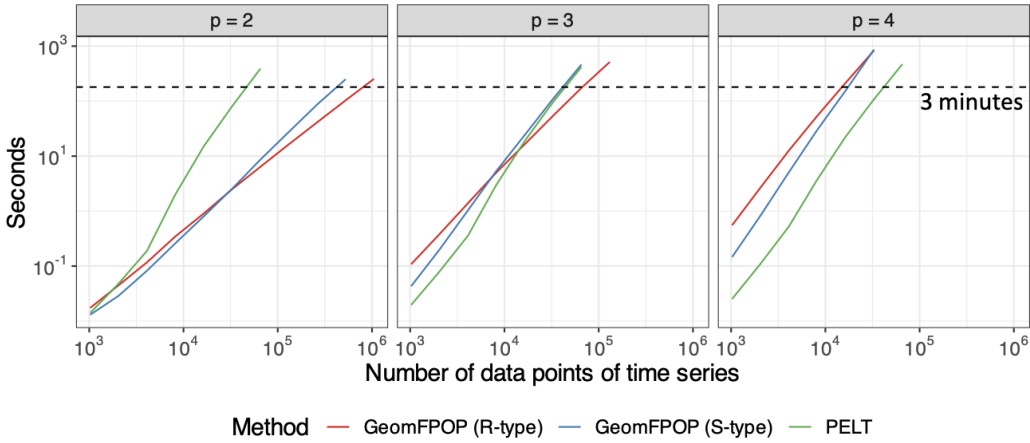

Figure 6: Run time of GeomFPOP (S and R types) and PELT using multivariate time series without change points. The maximum run time of the algorithms is 3 minutes. Averaged over 100 data sets.

## 4.3 Empirical Time Complexity of a Randomized GeomFPOP

R-type GeomFPOP is designed in such a way that at each iteration we need to consider all past and future spheres of change $i$. In practice, it is often sufficient to consider just a few of them to get an empty set. Having this in mind, we propose a further approximation of the $Z_t^i$ where we apply our $\bigcap_R$ and $\setminus_R$ operators only for a randomly selected subset of the past and future sets. In detail, we propose to redefine the output of the select() function in Algorithm 1 for any sets $\mathscr{P}^i$ and $\mathscr{F}^i(t)$ as:

- select($\mathscr{P}^i$) returns one random set from $\mathscr{P}^i$.
- select($\mathscr{F}^i(t)$) returns the last set $S_t^i$ and one random set from $\mathscr{F}^i(t)$.

Thus, we consider the following geometric update rule:

- (random/random) At time $t$ we update hyperrectangle:
    1. by only two intersection operations: one with the last S-type set $S_t^i$ from $\mathscr{F}^i(t)$, and one with a random S-type set from $\mathscr{F}^i(t)$;
    2. by only one exclusion operation with a random S-type set from $\mathscr{P}^i$.

In this approach, at time $t$ we need no more than three operations to update the testing set $\tilde{Z}_t^i$ for each $(i-1) \in \tau_t$. As can be seen in Figure Figure 11 of Section 5.5, by making less comparisons, we prune less change points than in the general GeomFPOP (R-type) case, but still more than PELT. It is important to note that in this randomization, we compare each change point candidate with only two other change point candidates (rather than all in the general case of GeomFPOP (R-type)). Therefore, informally our complexity at time step $t$ is only $\mathcal{O}(p|\tau_t^{GeomFPOP(\text{random}/\text{random})}|)$. According to the remark from Section 2.1 and the discussion in Section 4.2, even with large values of $p$, the overall complexity of GeomFPOP should not be worse than that of PELT. We investigated other randomized strategies (see Section 5.5) but this simple one was sufficient to significantly improve run times. The run time of our optimization approach and PELT in dimension ($p = 2, \ldots, 10, 100$) are presented in Figure 7. As in Section 4.2, run times were limited to three minutes and were recorded for simulations of length ranging $n$ from $2^{10}$ to $2^{23}$ data points ($p$-variate i.i.d. Gaussian noise).

Although the (random/random) approach reduces the quality of pruning (see Section 5.5), it gives a significant gain in run time compared to PELT in small dimensions. To be specific, with a run time of five minutes GeomFPOP, on average, processes a time series with a length of about $8 \times 10^6$, $10^6$ and $2, 5 \times 10^5$ data points in the dimensions $p = 2, 3$ and $4$, respectively. At the same time, PELT manages to process time series with a length of at most $6, 5 \times 10^4$ data points in these dimensions.

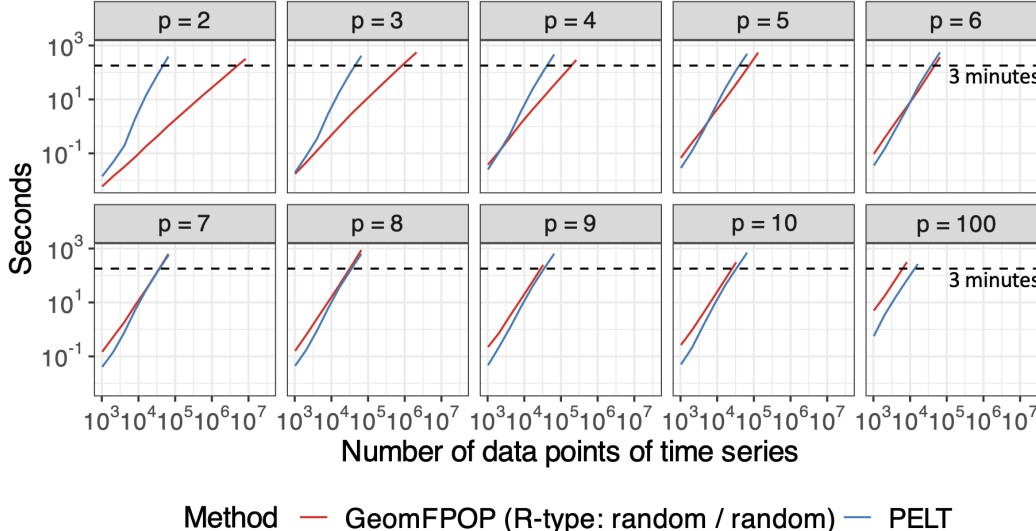

Figure 7: Run time of the (random/random) approach of { GeomFPOP} (R-type) and PELT using p-variate time series without change points ($p = 2, ..., 10, 100$). The maximum run time of the algorithms is 3 minutes. Averaged over 100 data sets.

## 4.4 Empirical Complexity of the Algorithm as a Function of $p$

We also evaluate the slope coefficient $\alpha$ of the run time curve of GeomFPOP with random sampling of the past and future candidates for all considered dimensions. In Figure 8 we can see that already for $p \geq 7$ $\alpha$ is close to 2.

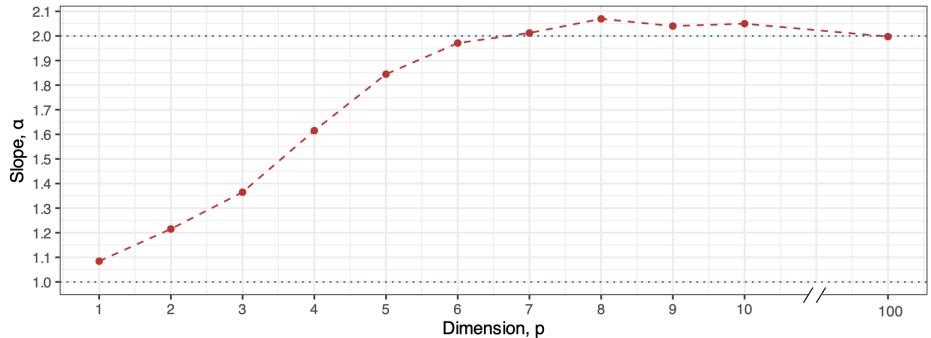

Figure 8: Run time dependence of (random/random) approach of GeomFPOP (R-type) on dimension $p$.

## 4.5 Run Time as a Function of the Number of Segments

For small dimensions we also generated time series with $n = 10^6$ data points with increasing number of segments. We have considered the following number of segments: $(1, 2, 5) \times 10^i$ (for $i = 0, ..., 3$) and $10^4$. The mean was equal to 1 for even segments, and 0 for odd segments. In Figure 9 we can see the run time dependence of the (random/random) approach of GeomFPOP (R-type) and PELT on the number of segments for this type of time series. For smaller number of segments (the threshold between small and large numbers of segments is around $5 \times 10^3$ for all considered dimensions $p$) GeomFPOP (random/random) is an order of magnitude faster. But for large number of segments, it can be seen that the run times (both PELT and GeomFPOP) are larger. This might be a bit counter-

intuitive. However, it is essential to recall that a similar trend of increased run time for a large number of segments was already noted in the one-dimensional case, as demonstrated in (Maidstone et al. 2017). This observation is explained as follows. When the number of segments becomes excessively large, the algorithm (both PELT and GeomFPOP) tends to interpret this abundance as an indication of no change, resulting in reduced pruning. As a conclusion of this simulation study, in Section 5.7 we make a similar analysis, but using time series in which changes are present only in a subset of dimensions. We observe that in this case GeomFPOP (random/random) will be slightly less effective but no worse than no change (see Figure 13).

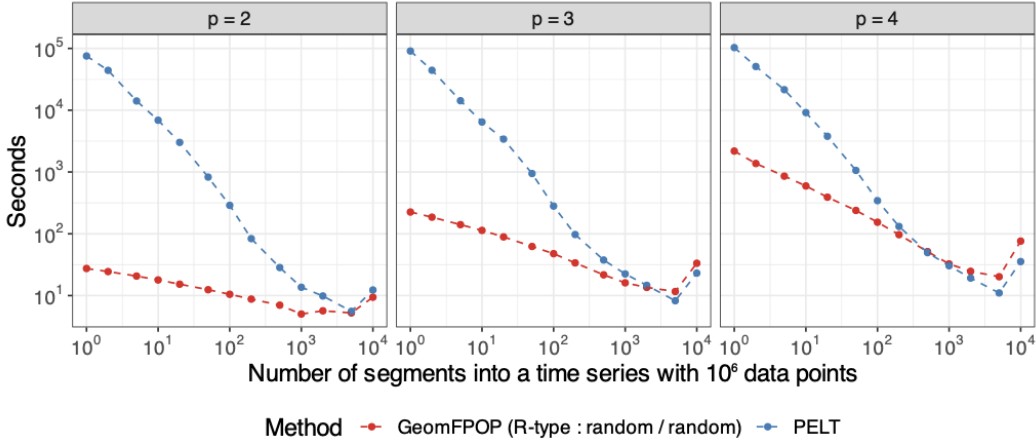

Figure 9: Run time dependence of (random/random) approach of GeomFPOP (R-type) on the number of segments in time series with $10^6$ data points.

## Acknowledgments

We thank Paul Fearnhead for fruitful discussions.

## 5 Supplements

### 5.1 Examples of Likelihood-Based Cost Functions

We define a cost function for segmentation as in Equation 1 by the function $\Omega(\cdot, \cdot)$ (the opposite log-likelihood (times two)). Below is the expression of this function linked to data point $y_i = (y_i^1, \ldots, y_i^p)$ in $\mathbb{R}^p$ for three examples of parametric multivariate models:

$$\Omega(\theta, y_i) = \begin{cases} \sum_{k=1}^{p} (y_i^k - \theta^k)^2, & \text{if } y_i \sim \mathcal{N}_p(\theta, \sigma^2 \mathbb{I}_p), \\ 2 \sum_{k=1}^{p} \left\{ \theta^k - \log\left( \frac{(\theta^k)^{y_i^k}}{y_i^k!} \right) \right\}, & \text{if } y_i \sim \mathcal{P}(\theta), \\ -2 \sum_{k=1}^{p} \log\left( (\theta^k)^{y_i^k} (1-\theta^k)^{\phi} \binom{y_i^k + \phi - 1}{y_i^k} \right), & \text{if } y_i \sim \mathcal{NB}(\theta, \phi). \end{cases} \quad (12)$$

We suppose that the over-dispersion parameter $\phi$ of the multivariate negative binomial distribution is known.

## 5.2 Intersection and Inclusion of Two p-balls

We define two $p$-balls, $S$ and $S'$ in $\mathbb{R}^p$ using their centers $c, c' \in \mathbb{R}^p$ and radius $R, R' \in \mathbb{R}^+$ as

$$S = \{x \in \mathbb{R}^p, \|x - c\|^2 \le R^2\} \text{ and } S' = \{x \in \mathbb{R}^p, \|x - c'\|^2 \le R'^2\},$$

where $\|x - c\|^2 = \sum_{k=1}^{p} (x^k - c^k)^2$, with $x = (x^1, ..., x^p) \in \mathbb{R}^p$, is the Euclidean norm. The distance between centers $c$ and $c'$ is defined as $d(c, c') = \sqrt{\|c - c'\|^2}$. We have the following simple results:

$$S \cap S' = \varnothing \iff d(c, c') > R + R',$$

$$S \subset S' \text{ or } S' \subset S \iff d(c, c') \le |R - R'|.$$

## 5.3 Intersection and Inclusion Tests

*Remark.* For any $S_j^i \in \mathbf{S}$ its associated function $s$ can be redefine after normalization by constant $j - i + 1$ as:

$$s(\theta) = a(\theta) + \langle b, \theta \rangle + c,$$

with $a(\cdot)$ is some convex function depending on $\theta$, $b = \{b^k\}_{k=1,...,p} \in \mathbb{R}^p$ and $c \in \mathbb{R}$.

For example, in the Gaussian case, the elements have the following form:

$$a : \theta \mapsto \theta^2, \quad b^k = 2\bar{Y}_{i:j}^k, \quad c = \bar{Y}_{i:j}^2 - \Delta_{ij},$$

where $\bar{Y}_{i:j}^k = \frac{1}{j-i+1} \sum_{u=i+1}^{j} y_u^k$ and $\bar{Y}_{i:j}^2 = \frac{1}{j-i+1} \sum_{u=i+1}^{j} \sum_{k=1}^{p} (y_u^k)^2$.

**Definition 5.1.** For all $\theta \in \mathbb{R}^p$ and $S_1, S_2 \in \mathbf{S}$ with their associated functions, $s_1$ and $s_2$, we define a function $h_{12}$ and a hyperplane $H_{12}$ as:

$$h_{12}(\theta) := s_2(\theta) - s_1(\theta), \quad H_{12} := \{\theta \in \mathbb{R}^p | h_{12}(\theta) = 0\}.$$

We denote by $H_{12}^+ := \{\theta \in \mathbb{R}^p | h_{12}(\theta) > 0\}$ and $H_{12}^- := \{\theta \in \mathbb{R}^p | h_{12}(\theta) < 0\}$ the positive and negative half-spaces of $H_{12}$, respectively. We call $\mathbf{H}$ the set of hyperplanes.

For all $S \in \mathbf{S}$ and $H \in \mathbf{H}$ we introduce a $\mathtt{half-space}$ operator.

**Definition 5.2.** The operator $\mathtt{half-space}$ is such that:

1. the left input is an S-type set $S$;
2. the right input is a hyperplane $H$;
3. the output is the half-spaces of $H$, such that $S$ lies in those half-spaces.

**Definition 5.3.** We define the output of $\mathtt{half-space}(S, H)$ by the following rule:

1. We find two points, $\theta_1, \theta_2 \in \mathbb{R}^p$, as:

$$
\begin{cases}
\theta_1 = & Arg\min s(\theta), \\
\theta_2 = & \begin{cases} Arg\min\limits_{\theta \in S} h(\theta), & \text{if } \theta_1 \in H^+, \\ Arg\max\limits_{\theta \in S} h(\theta), & \text{if } \theta_1 \in H^-. \end{cases}
\end{cases}
$$

2. We have:

$$
\mathtt{half-space}(S, H) = \begin{cases}
\{H^+\}, & \text{if } \theta_1, \theta_2 \in H^+, \\
\{H^-\}, & \text{if } \theta_1, \theta_2 \in H^-, \\
\{H^+, H^-\}, & \text{otherwise.}
\end{cases}
$$

**Lemma 5.1.** $S_1 \subset H_{12}^- \Leftrightarrow \partial S_1 \subset H_{12}^-$, where $\partial(\cdot)$ denote the frontier operator.

The proof of Lemma 5.1 follows from the convexity of $S_1$.

**Lemma 5.2.** $S_1 \subset S_2$ (resp. $S_2 \subset S_1$) $\Leftrightarrow S_1, S_2 \subset H_{12}^-$ (resp. $S_1, S_2 \subset H_{12}^+$).

*Proof.* We have the hypothesis $\mathcal{H}_0 : \{S_1 \subset S_2\}$, then

$$
\forall \theta \in \partial S_1 \quad \begin{cases} s_1(\theta) = 0, & \text{[by Definition 1.1]} \\ s_2(\theta) \leq 0, & \text{[by } \mathcal{H}_0\text{]} \end{cases} \Rightarrow \theta \in H_{12}^- \quad \Rightarrow \partial S_1 \subset H_{12}^-.
$$

Thus, according to Lemma 5.1, $S_1 \subset H_{12}^-$.

We have now the hypothesis $\mathcal{H}_0 : \{S_1, S_2 \subset H_{12}^-\}$, then

$$
\forall \theta \in S_1 \quad \begin{cases} s_1(\theta) \leq 0, & \text{[by Definition 1.1]} \\ h_{12}(\theta) < 0, & \text{[by } \mathcal{H}_0, \text{ Definitions 5.1 and 1.1]} \end{cases} \Rightarrow \theta \in S_2 \quad \Rightarrow S_1 \subset S_2.
$$

Similarly, it is easy to show that $S_2 \subset S_1 \Leftrightarrow S_1, S_2 \subset H_{12}^+$.

$\square$

**Lemma 5.3.** $S_1 \cap S_2 = \varnothing \Leftrightarrow H_{12}$ is a separating hyperplane of $S_1$ and $S_2$.

*Proof.* We have the hypothesis $\mathcal{H}_0 : \{S_1 \subset H_{12}^+, S_2 \subset H_{12}^-\}$. Thus, $H_{12}$ is a separating hyperplane of $S_1$ and $S_2$ then, according to its definition, $S_1 \cap S_2 = \varnothing$.

We have now the hypothesis $\mathcal{H}_0 : \{S_1 \cap S_2 = \varnothing\}$ then

$$
\forall \theta \in S_1 \quad \begin{cases} s_1(\theta) \leq 0, & \text{[by Definition 1.1]} \\ s_2(\theta) > 0, & \text{[by } \mathcal{H}_0, \text{ Definition 1.1]} \end{cases} \Rightarrow \theta \in H_{12}^+.
$$

$$
\forall \theta \in S_2 \quad \begin{cases} s_1(\theta) > 0, & \text{[by } \mathcal{H}_0, \text{ Definition 1.1]} \\ s_2(\theta) \leq 0, & \text{[by Definition 1.1]} \end{cases} \Rightarrow \theta \in H_{12}^-.
$$

Consequently, $H_{12}$ is a separating hyperplane of $S_1$ and $S_2$.

$\square$

**Proposition 5.1.** *To detect set inclusion $S_1 \subset S_2$ and emptiness of set intersection $S_1 \cap S_2$, it is necessary:*

1. *build the hyperplane $H_{12}$;*
2. *apply the* half − space *operator for couples $(S_1, H_{12})$ and $(S_2, H_{12})$ to know in which half-space(s) $S_1$ and $S_2$ are located;*
3. *check the conditions in Lemmas 5.2 and 5.3.*

## 5.4 Proof of Proposition 3.2

*Proof.* Let $\mathbf{c} = \{\mathbf{c}^k\}_{k=1,\dots,p}$ is the minimal point of $S$, defined as in Equation 10. In the intersection case, we consider solving the optimization problem (9) for the boundaries $\tilde{l}^k$ and $\tilde{r}^k$, removing constraint $l^k \leq \theta^k \leq r^k$. If $R$ intersects $S$, the optimal solution $\theta^k$ belongs to the boundary of $S$ due to our simple (axis-aligned rectangular) inequality constraints and we get

$$s^k(\theta^k) = -\sum_{j \neq k} s^j(\theta^j) + \Delta. \tag{13}$$

We are looking for minimum and maximum values in $\theta^k$ for this equation with constraints $l^j \leq \theta^j \leq r^j$ ($j \neq k$). Using the convexity of $s^k$ and $s^j$, we need to maximize the quantity in the right-hand side. Thus, the solution $\tilde{\theta}^j$ for each $\theta^j$ is the minimal value of $\sum_{j \neq k} s^j(\theta^j)$ under constraint $l^j \leq \theta^j \leq r^j$ and the result can only be $l^j$, $r^j$ or $\mathbf{c}^j$. Looking at all coordinates at the same time, the values for $\tilde{\theta} \in \mathbb{R}^p$ corresponds to the closest point $\mathbf{m} = \{\mathbf{m}^k\}_{k=1,\dots,p}$. Having found $\theta^{k_1}$ and $\theta^{k_2}$ using $\tilde{\theta}$ the result in Equation 11 is obvious considering current boundaries $l^k$ and $r^k$.

In exclusion case, we remove from $R$ the biggest possible rectangle included into $S \cap \{l^j \leq \theta^j \leq r^j, j \neq k\}$, which correspond to minimizing the right hand side of Equation 13, that is maximizing $\sum_{j \neq k} s^j(\theta^j)$ under constraint $l^j \leq \theta^j \leq r^j$ ($j \neq k$). In that case, the values for $\tilde{\theta}$ correspond to the greatest value returned by $\sum_{j \neq k} s^j(\theta^j)$ on interval boundaries. With convex functions $s^j$, it corresponds to the farthest point $\mathbf{M} = \{\mathbf{M}^k\}_{k=1,\dots,p}$.

$\square$

## 5.5 Optimization Strategies for GeomFPOP (R-type)

In GeomFPOP(R-type) at each iteration, we need to consider all past and future spheres of change $i$. As it was said in Section 4, in practice it is often sufficient to consider just a few of them to get an empty set. Thus, we propose to limit the number of operations $\cap_R$ no more than two:

- last. At time $t$ we update hyperrectangle by only one operation, this is an intersection with the last S-type set $S_t^i$ from $\mathscr{F}^i(t)$.
- random. At time $t$ we update the hyperrectangle by only two operations. First, this is an intersection with the last S-type set $S_t^i$ from $\mathscr{F}^i(t)$, and second, this is an intersection with other random S-type set from $\mathscr{F}^i(t)$.

The number of operations $\setminus_R$ we limit no more than one:

- empty. At time $t$ we do not perform $\setminus_R$ operations.
- random. At time $t$ we update hyperrectangle by only one operation: exclusion with a random S-type set from $\mathscr{P}^i$.

According to these notations, the approach presented in the original GeomFPOP (R-type) has the form (all/all). We show the impact of introduced limits on the number of change point candidates retained over time and evaluate their run times. The results are presented in Figures 10 and 11.

Even though the (random/random) approach reduces the quality of pruning in dimensions $p = 2, 3$ and 4, it gives a significant gain in the run time compared to the original GeomFPOP (R-type) and is at least comparable to the (last/random) approach.

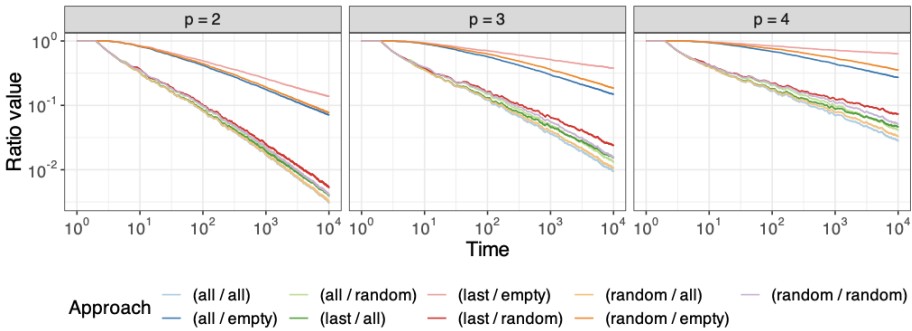

Figure 10: Ratio number of candidate change point over time by different optimization approaches of GeomFPOP (R-type) in dimension $p = 2, 3$ and 4. Averaged over 100 data sets without changes with $10^4$ data points.

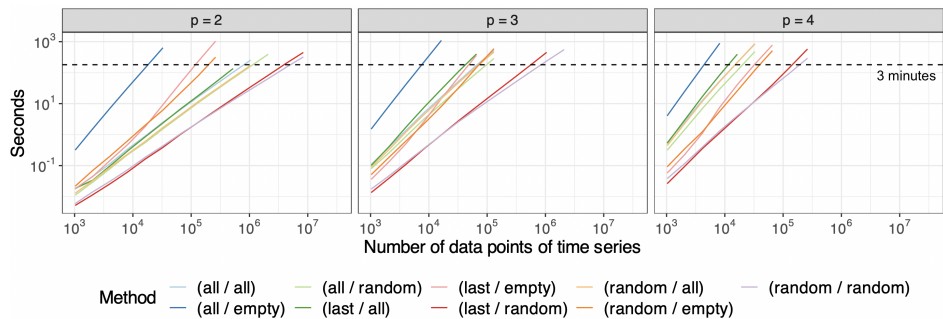

Figure 11: Run time of different optimization approaches of GeomFPOP (R-type) using multivariate time series without change points. The maximum run time of the algorithms is 3 minutes. Averaged over 100 data sets.

## 5.6 The number of change point candidates in time: GeomFPOP vs. PELT

In this appendix we compare the square of the number of candidates stored by GeomFPOP (S and R-type) to the corresponding number of candidates stored by PELT over time. Indeed, the complexity of GeomFPOP at each time step is a function of the square of the number of candidates, while, for PELT, of the number of candidates (see Section 4.2). Figure 12 shows the ratios of these computed quantities for dimension $2 \le p \le 10$. It is noteworthy that for both S-type and R-type for $p = 2$ this ratio is almost always less than 1 and decreases with time. This is coherent with the fact that GeomFPOP is faster than PELT (see Figure 6). At $p = 3$ for the R-type this ratio is approximately 1, while for the S-type it is greater than 1 and continues to increase with increasing $t$ value. For sizes $3 < p \le 10$, this ratio remains consistently greater than 1 for both S-type and R-type, showing a continuous increasing trend with time. This is coherent with the fact that GeomFPOP is almost as fast as PELT for $p = 3$ and slower than PELT for $p \le 4$ (see Figure 6).

## 5.7 Run time of the algorithm by multivariate time series with changes in subset of dimension

We expect GeomFPOP (random/random) to be slightly less effective (but no worse than in the absence of changes) if changes are only present in a subset of dimensions. To this end, in this appendix for

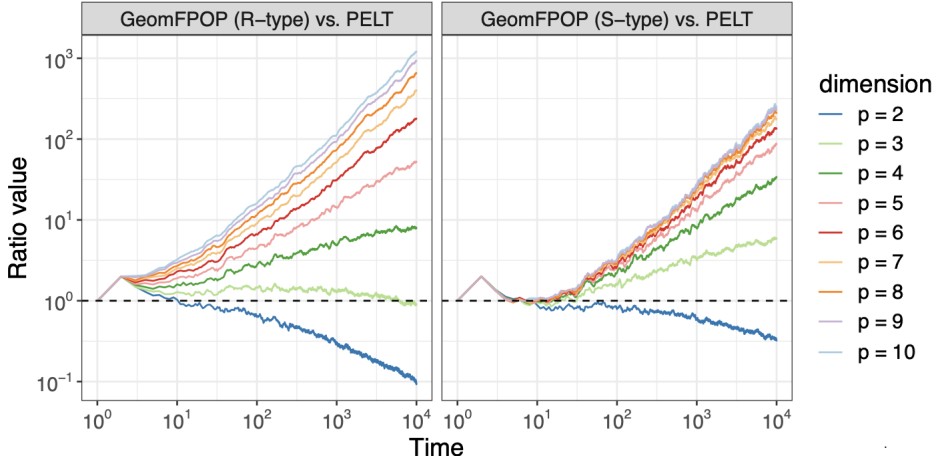

Figure 12: The ratio of the square of the number of candidates stored in GeomFPOP to the number of candidates stored in PELT over time. The horizontal black line corresponds to the value 1.

dimension $2 \leq p \leq 4$ we examine the run time of GeomFPOP (`random/random`) as in Section 4.5 (see Figure 9) but removing all changes in the last $k$ dimensions (with $k = 0, \ldots, p - 1$). The results are presented in Figure 13. There are two regimes. For a small number of segments (the threshold between small and large numbers of segments is around $2 \times 10^3$ for all considered dimensions $p$), the run time decreases with the number of segments and the difference between the run time of GeomFPOP (`random/random`) for $k = 0$ (this case corresponds to changes in all dimensions) and $k > 0$ is very small. For larger number of segments, the run time increases with the number of segments, as in Section 4.5 and also increases with $k$. Importantly, in this regime the run time is never lower than for 1 segment.

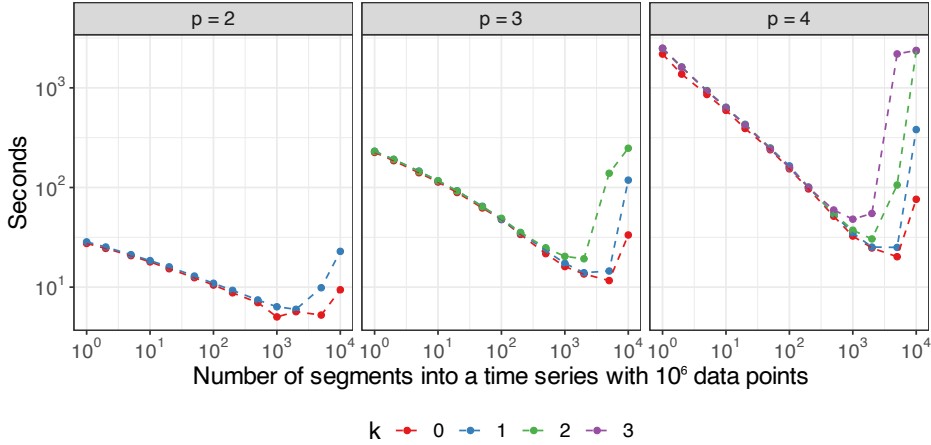

Figure 13: Dependence of the run time of the (`random/random`) approach of GeomFPOP (R-type) on the number of segments in a $p$-variable time series with $10^6$ data points where all changes in the last $k$ dimensions have been removed.

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
