# OpenReview forum: "Geometric-Based Pruning Rules For Change Point Detection in Multiple Independent Time Series."
_Computo — Accepted by Computo_

### Review · Reviewer_eaEU · 2024-03-21

**Summary Of Contributions:**

Having read the responses to all the reviewers and the new material provided by the authors (new paragraphs and new simulations), I believe they have convincingly addressed all our concerns. I therefore suggest that we accept publication of the article in its present form.

**Audience:**

Yes

**Claims And Evidence:**

Yes

**Requested Changes:**

Having read the responses to all the reviewers and the new material provided by the authors (new paragraphs and new simulations), I believe they have convincingly addressed all our concerns. I therefore suggest that we accept publication of the article in its present form.

**Strengths And Weaknesses:**

N/A

---

> ### Author Response · Authors · 2024-05-17
> **Second revision of article "Geometric-Based Pruning Rules For Change Point Detection in Multiple Independent Time Series".**
>
> Dear Reviewer,
>
> Thank you for your review of our work and for your constructive comments. We are pleased that our responses and changes have addressed your comments and concerns. Your valuable recommendations helped significantly improve the quality of the article. In the updated version (2 revision) of GeomFPOP, you can familiarize with the version of the article after making the changes suggested by Reviewer 3.
>
> Best regards, Liudmila Pishchagina, Vincent Runge and Guillem Rigaill.

---

### Review · Reviewer_V5Zo · 2024-05-06

**Summary Of Contributions:**

As saying in the first review, the work makes an interesting and important contribution to the computational aspect for segmentation problems.

**Audience:**

Yes

**Broader Impact Concerns:**

no ethical implications of the work that would require the addition of a broader impact statement

**Claims And Evidence:**

Yes

**Requested Changes:**

The manuscript has been substantially improved. I have minor comments or suggestions for further refinement:

** In section 1.2 at the beginning of page 6:  ``It essentially relies on the assumption that splitting a segment in two ....''. This is always true not specific to PELT? The penalty constant $\beta$ is missing in the expression on the left?

** I suggest ``bivariate'' instead of ``bi-variate'' (page 41 of the response, the authors said that it has been edited but no correction has been made) and ``multivariate'' instead of ``multi-variate''

** my comment on line 2 of page 8 is in the paragraph {\textit {Inequality-based pruning geometry}}: this is $S_t^i$ or rather $S_{t+}^i$ ?

**Strengths And Weaknesses:**

I have no weaker elements in this new version of the article. The authors have responded perfectly to my requests, comments and suggestions.

---

> ### Author Response · Authors · 2024-05-17
> **Second revision of article "Geometric-Based Pruning Rules For Change Point Detection in Multiple Independent Time Series". Answers to the questions of the Reviewer 3**
>
> Dear Reviewer 3,
>
> Thank you for your positive feedback and careful consideration of our work. We are glad that the changes we made satisfied your requests and comments.
>
> Below we provide answers to your questions.
>
>
> Question 1 : In section 1.2 at the beginning of page 6: ``It essentially relies on the assumption that splitting a segment in two ....''. This is always true not specific to PELT? The penalty constant $\beta$ is missing in the expression on the left?
>
> Answer: Thanks for this question.
>
> This assumption is true assuming parameters of different segments are independent. It is typically false otherwise (e.g.,  P. Fearnhead, R. Maidstone, and A. N. Letchford. Detecting changes in slope with an l0 penalty, 2017 ; V. Runge, T. D. Hocking, G. Romano, F. Afghah, P. Fearnhead, and G. Rigaill. gfpop: an r package for univariate graph-constrained change-point detection, 2022).
>
> The penalty for the number of changes, $\beta$ , is not included in the PELT rule: the term $\beta$ is not missing.
>
>
> Question 2 : I suggest bivariate'' instead of bi-variate'' (page 41 of the response, the authors said that it has been edited but no correction has been made) and multivariate'' instead of multi-variate''.
>
> Answer: We apologize for this confusion. These changes have now been made to the manuscript.
>
>
> Question 3 : my comment on line 2 of page 8 is in the paragraph {\textit {Inequality-based pruning geometry}}: this is $S_t^i$ or rather $S_{t+}^i$ ?
>
> Answer: Thanks for this clarification. Indeed, in paragraph Inequality based pruning geometry (and also in paragraph Functional pruning geometry) there is this error, to eliminate it we replaced everywhere in these two paragraphs the moment of time t+1 with t.
>
>
> Best regards,
> Liudmila Pishchagina, Vincent Runge and Guillem Rigaill.

---

### Comment · Reviewer_eaEU · 2024-03-14
**Rebuttal**

Given that this article is a revision carried out after a first review round, would it be possible for the authors to provide a detailed "point by point" response to the reviewers and/or highlight in the manuscript the changes that have been made from the original manuscript?

---

> ### Author Response · Authors · 2024-03-14
> **A detailed response to the reviewers  in file "GeomFPOP_for reviewers_Resubmission.pdf" (see 28-44 pages).**
>
> Hello,
>
> Indeed, this article underwent revision after the initial review round. In the Supplementary Material (.zip) file "GeomFPOP_for_reviewers_Resubmission.pdf" (pages 1-27), we highlight the changes made to the manuscript from its original version:
>
> Original text is displayed in gray.
>
> Changes made after reviewer 1's questions are indicated in brown.
>
> Changes made after reviewer 2's questions are marked in orange.
>
> Changes made after reviewer 3's questions are shown in blue.
>
> Additionally, we have provided a detailed "point-by-point" response to the reviewers in the same file, found on pages 28-44.
>
> In the "5 NEW_ADDITIONS to the article" section of the Supplementary Material (.zip), we have included new simulations (R code and results), which have been incorporated into the article.
>
> We apologize for any inconvenience this may have caused. Please let me know if you were able to access the file.
>
> Regards,
> Liudmila Pishchagina

---

### Note · Reviewer_eaEU · 2024-05-21

**Audience:**

Yes

**Claims And Evidence:**

Yes

**Decision Recommendation:**

Accept

---

### Note · Reviewer_V5Zo · 2024-06-03

**Comment:**

In the name of reviewer V7zo

The manuscript has been substantially improved. I have minor comments or suggestions for further refinement:

** In section 1.2 at the beginning of page 6: ``It essentially relies on the assumption that splitting a segment in two ....''. This is always true not specific to PELT? The penalty constant $\beta$ is missing in the expression on the left?

** I suggest bivariate'' instead of bi-variate'' (page 41 of the response, the authors said that it has been edited but no correction has been made) and multivariate'' instead of multi-variate''

** my comment on line 2 of page 8 is in the paragraph {\textit {Inequality-based pruning geometry}}: this is $S_t^i$ or rather $S_{t+}^i$ ?

**Audience:**

Yes

**Claims And Evidence:**

Yes

**Decision Recommendation:**

Accept

---

> ### Comment · Reviewer_V5Zo · 2024-06-05
>
> The authors have responded perfectly to my last three suggestions.

---

### Note · Action_Editor_WZbZ · 2024-06-03

**Comment:**

Please amend according to the minor modifications raised by the reviewers before final acceptance.

**Audience:**

Yes

**Claims And Evidence:**

Yes

**Decision Recommendation:**

Accept

---

### Decision · Action_Editor_WZbZ · 2024-06-03

**Recommendation:** Accept as is

**Comment:**

The authors have addressed all the points raised by the reviewers.

**Audience:**

This paper would be of interest for researcher or practitioner facing a segmentation problem.

**Claims And Evidence:**

Dear authors,

I am pleased to inform you that your paper is accepted pending minor revisions requested by the reviewers.

---

> ### Decision · Editors_In_Chief · 2024-06-21
>
> I approve the AE's decision.